# Antimicrobial Activity of *Myrtus communis* L. and *Rosmarinus officinalis* L. Essential Oils against *Listeria monocytogenes* in Cheese

**DOI:** 10.3390/foods10051106

**Published:** 2021-05-17

**Authors:** Cristina Saraiva, Ana Catarina Silva, Juan García-Díez, Beniamino Cenci-Goga, Luca Grispoldi, Aníbal Filipe Silva, José Manuel Almeida

**Affiliations:** 1Veterinary and Animal Research Centre (CECAV), University of Trás-os-Montes e Alto Douro, 5001-801 Vila Real, Portugal; catarinasilvaa_13@hotmail.com (A.C.S.); juangarciadiez@gmail.com (J.G.-D.); 2Department of Veterinary Sciences, School of Agrarian and Veterinary Sciences, University of Trás-os-Montes e Alto Douro, 5000-801 Vila Real, Portugal; 3Medicina Veterinaria, Laboratorio di Ispezione degli Alimenti di Origine Animale, Università degli Studi di Perugia, 06126 Perugia, Italy; beniamino.cencigoga@unipg.it (B.C.-G.); grisluca@outlook.it (L.G.); 4Department of Paraclinical Sciences, Faculty of Veterinary Science, University of Pretoria, Onderstepoort 0110, South Africa; 5Centre of Applied Photonics, INESC-TEC, Faculty of Sciences of the University of Porto, Rua do Campo Alegre, s/n, 4169-007 Porto, Portugal; filipe.m.silva@inesctec.pt (A.F.S.); jose.almeida@inesctec.pt (J.M.A.); 6Department of Physics, School of Sciences and Technology, University of Trás-os-Montes e Alto Douro, 5001-801 Vila Real, Portugal

**Keywords:** cheese, essential oils, *Listeria monocytogenes*, *Myrtus* *communis* L., *Rosmarinus* *officinalis* L., food safety

## Abstract

*Listeria monocytogenes* has been referred to as a concern microorganism in cheese making due to its ability to survive and grow in a wide range of environmental conditions, such as refrigeration temperatures, low pH and high salt concentration at the end of the production process. Since cheese may be a potential hazard for consumers, especially high-risk consumers (e.g., pregnant, young children, the elderly, people with medical conditions), efforts of the dairy industry have been aimed at investigating new conservation techniques based on natural additives to meet consumers’ demands on less processed foods without compromising the food safety. Thus, the aim of this study was to evaluate the efficacy of *Myrtus communis* L. (myrtle) and *Rosmarinus officinalis* L. (rosemary) essential oils (EO) against *Listeria monocytogenes* ATCC 679 spiked in sheep cheese before ripening. After the cheesemaking process, the samples were stored at 8 °C for 2 h, 1 d, 3 d, 14 d and 28 d. The composition of EO was identified by gas chromatography-mass spectrometry (GC-MS) analysis. Constituents such as 1,8-cineole, limonene, methyl-eugenol, α-pinene, α-terpineol, α-terpinolene and β-pinene were present in both EO, accounting for 44.61% and 39.76% from the total of chemical compounds identified for myrtle and rosemary EO, respectively. According to the chemical classification, both EO were mainly composed of monoterpenes. Minimum inhibitory concentration (MIC) against *L. monocytogenes* was obtained at 31.25 μL/mL to myrtle EO and at 0.40 μL/mL to rosemary EO. Then, cheeses were inoculated with *L. monocytogenes* (Ca. 6 log CFU/mL) and EO was added at MIC value. The addition of rosemary and myrtle EO displayed lower counts of *L. monocytogenes* (*p* < 0.01) (about 1–2 log CFU/g) during the ripening period compared to control samples. Ripening only influences (*p* < 0.001) the growth of *L. monocytogenes* in control samples. Since rosemary and myrtle EO do not exert any negative impact on the growth of native microflora (*p* > 0.05), their use as natural antimicrobial additives in cheese demonstrated a potential for dairy processors to assure safety against *L. monocytogenes.*

## 1. Introduction

*Listeria monocytogenes* is an important foodborne pathogen with relevant importance in dairy products such as milk and soft cheese [1]. This pathogen is not nutritionally demanding and can grow in a wide range of environmental conditions such as low pH, high water activity (a_w_) values and refrigeration temperatures [2,3]. Detection of *L. monocytogenes* in several dairy industries denotes its global distribution as well as the endemic trait of this microorganism. Since *L. monocytogenes* can be present in raw milk, its entrance into the cheese manufacturing process may compromise its safety with special relevance in fresh cheese and those made from raw milk [4]. Although pasteurization treatment increases the safety of cheese, foodborne outbreaks still occur [5]. Cross-contamination during the cheesemaking process has been described as a source of potential cheese contamination, which may occur at several different stages of the production and from multiple origins such as ingredients, food handlers’ manipulation or surfaces among others [6]. Therefore, due to the ubiquitous nature of *L. monocytogenes*, proper cleaning and disinfection procedures are necessary to avoid the formation of biofilms. Consequently, an adequate and validated cleaning and disinfection pre-requisite program and HACCP is essential as indicated by the European guidelines to comply with the food regulations [7]. 

Despite these measures, cheese is still a food susceptible to contamination by pathogenic and spoilage microorganisms, which lead from reduced commercial validity to serious risks to the consumer’s health [8]. To improve the safety of cheese, some techniques such as thermal processing, high hydrostatic pressure, high voltage atmospheric cold plasma, several types of packaging as well as the use of active compounds such as chitosan, bacteriocins or essential oils have been studied to control foodborne and spoilage bacteria [9,10,11,12,13].

Regarding essential oils (EO), their use has received particular attention from the dairy industry since they represent a natural and economic source of natural antimicrobials for food safety and shelf-life extension [8]. Congruently, the use of EO enhances the greenlabelling and fosters a natural image of cheese, avoiding the negative perception of consumers about the use of synthetic chemical preservatives [14]. However, specific characteristics of cheeses, as well as extrinsic factors, can influence the antimicrobial effect of EO when incorporated into the cheese. It indicates that antimicrobial properties of EO must be assessed both in vitro and in real food products [15]. In addition, the use of EO may impact the sensory properties of cheeses [16]. Since the food industry requires the development of new strategies to comply with the consumers´ requirements, without compromising the safety and quality of foodstuffs [1], the present work aims to evaluate the antimicrobial effect of myrtle and rosemary EO against *L. monocytogenes* in fresh cheese during the ripening process and the impact of EO on cheese natural microbiota.

## 2. Materials and Methods

### 2.1. Preparation of Plant Material

Fresh aerial parts of myrtle (*Myrtus communis* L.) and rosemary (*Rosmarinus officinalis* L.) were collected from September to December in Northern Portugal. Specimens were identified by dichotomous key. Samples of fresh leaves were washed with distilled water, dried until stable weight and stored in plastic bags at room temperature until use. 

### 2.2. Extraction of Essential Oils

EO were extracted from dried leaves of each plant by steam distillation in a Clevenger-type apparatus. The biomass (30 g of dried leaves) was placed into the round-bottom flask to which 350 mL of distilled water was added, heated by electric resistance. Samples were subjected to hydrodistillation for 3–4 h at constant temperature (105 °C). The EO obtained were stored in sealed vials at 3–4 °C, in the absence of light.

### 2.3. Gas Chromatography-Mass Spectrometry (GC-MS) Analysis

The volatile profile of EO was analyzed by GC-MS using a Thermo Scientific™ TRACE™ 1300 Gas Chromatograph combined with an ISQ™ Series Single Quadrupole MS (Thermo Fisher Scientific Inc., Waltham, MA, USA). The separation of volatile compounds was performed on a Thermo Scientific TG-5MS column (60 m × 0.25 mm × 0.25 µm). A temperature gradient was programmed, starting from 60 °C/2 min to to 280 °C in a gradient of 10 °C/min, and held for 5 min. Samples and standards were prepared prior to analysis using n-hexane (Merck) in 1.0% (*v*/*v*) and 0.2% (*v*/*v*) concentrations, respectively, and the volume injected was 1.0 µL, using an autosampler. The injector was set to split mode (1:5), operating at 250 °C and 165 KPa. The transfer line temperature and ion source were set to 280 °C and 250 °C, respectively, with the last operating under electron impact mode (70 eV, mass scan range 30–400 amu). All analytical separations were performed using Helium 99.999% as carrier gas. Identification of volatile compounds was performed using NIST/EPA/NIH Mass Spectral Library (2011) and other libraries, namely Pherobase, as well as by comparison of authentic standards or data from the literature.

### 2.4. Inoculum Preparation 

*L. monocytogenes* strain ATCC 679 was cultured in tryptone soya broth (Oxoid, Hampshire, UK) at 30 °C for 18 h. Culture was then transferred to a sterile centrifuge bottle and centrifuged at 10,000× *g* for 10 min at 4 °C. The supernatant was decanted and the pellet was suspended in sterile 0.1% peptone solution. The washing step was repeated twice. The turbidity of the suspension was adjusted by optical density (O.D.) at 600 nm (Ca. 8 log CFU/mL). Serial (10-fold) dilutions were made in NaCl 0.85% to achieve the required inoculation level (Ca. 6 log CFU/mL). Cultures of *L. monocytogenes* were plated in Oxford Agar (Biokar Diagnostics BK110, Beauvais, France) in triplicate at 37 °C for 48 h to verify the number of viable *L. monocytogenes* in the suspension.

### 2.5. Microtiter Plate Assay (MPA)

The minimum inhibitory concentration (MIC) and minimal bactericidal concentration (MBC) were studied for myrtle and rosemary EO. MIC was determined using a broth microdilution. Bacterial inoculums were prepared from 24 h broth cultures and suspensions were adjusted to 0.5 McFarland standard. EO dilutions were prepared directly on the Mueller–Hinton broth (MHB, Biokar, Beauvais, France) to double the desired final concentration. The inoculum of the target microorganism was prepared also in MHB to double the designed concentration (ca. 6 log CFU/mL to result in half of that concentration in the well). As defined in previous studies [17], geometric and successive dilutions of EO were performed in a sterile 96-well microtiter plate. In each well, 100 μL of MHB with each EO dilution was added to 20 µL of the resazurin aqueous solution (135 mg in 40 mL sterile distilled water) and 100 μL of MHB with the bacterial inoculum. Positive control (presence of *L. monocytogenes* and absence of EOs) and negative controls (addition of EO and absence of *L. monocytogenes*) were performed with 100 μL MHB to which 20 μL of the aqueous resazurin solution was added. Afterwards, the plates were covered, incubated at 37 °C for 24 h and monitored for visible growth. The MIC was considered the lowest concentration of EO at which bacteria failed to grow, detected by the unaided eye, matching with the negative control. In those cases in which ambiguous turbidity was detected, the test was complemented with the seeding of a 10 μL loop in Mueller–Hinton Agar (MHA, Biokar, Beauvais, France) to confirm the absence of growth. To evaluate the MBC, 10 μL of each assay, in which no microbial growth was observed, was spread into MHA and was incubated for 24 h. The MBC was considered as the lowest concentration determining a reduction in the population of 4 log.

### 2.6. Microbiological Challenge Test

#### 2.6.1. Cheese Production and Inoculation

Raw sheep’s milk supplied from an artisanal company (from the Serra da Estrela region of Portugal) was used in the manufacture of cured mini-cheeses, with a semi-soft paste, produced by the traditional method. The manufacture starts with milk heating, in a water bath, up to a temperature of 35 °C. Then, liquid rennet (1/4000) (ABIASA) was added to coagulate the milk and homogenization was carried out. Total coagulation was achieved in 90 min. Afterwards, the coagulum was cut into 1–1.5 cm cubes, leaving it to stand for 2 min for draining to take place, with the assistance of a cloth. Draining was performed in about 25–30 min. After this stage, molding was carried out, in which the cubes were placed into cylindrical strainers with a capacity for 40 g of cheese that allows (upon force exertion) the serum excess to be drained out. Then, salting was carried out, by placing the cheeses (immersion) in a salt-water with a solution of water with NaCl at a concentration of 0.20 g/cm^3^, for 10 min on each side. 

Then, fresh mini-cheeses (40 g) were separated into two batches: (i) cheese contaminated with 0.08 mL of bacterial suspension of *L. monocytogenes* and (ii) control cheeses without contamination. Contaminated batches with *L. monocytogenes* were divided into 2 groups: (i) cheeses made with rosemary EO and (ii) cheeses made with myrtle EO. Both contaminated batches were made with 0.02 mL of each EO (myrtle or rosemary) added at MIC values. Control batches were divided into 3 groups: (i) absence of listerial contamination and absence of EO in its composition, (ii) addition of rosemary (0.02 mL at MIC concentration) EO without *L. monocytogenes* contamination and iii) addition of myrtle EO (0.02 mL at MIC concentration) without *L. monocytogenes* contamination.

Subsequently, for ripening, cheeses were properly placed in a refrigeration chamber with controlled conditions (temperature at 8 °C and 90% of relative humidity), with variable storage periods (2 h, 12 h, 1 d, 3 d, 6 d, 13 d and 28 d). Control samples were removed from storage for physical–chemical (pH, a_w_) and microbiological (total viable counts at 30 °C, lactic acid bacteria [18], and moulds and yeasts enumeration [19]) analysis. Control and inoculated samples were also enumerated for *L. monocytogenes* [20]. Microbiological analysis for each sample was carried out in triplicate.

#### 2.6.2. Physical–Chemical Analysis

The pH was measured directly in the cheese using a pH meter (Crison, Barcelona, Spain) with a penetration probe (Mettler-Toledo, Giesen, Germany). The a_w_ was measured in a Hygroscope DT apparatus (Rotronic, Zurich, Switzerland) with a WA40 cell maintained at 20 ± 2 °C.

#### 2.6.3. Microbial Analysis

Ten grams of each fresh cheese was aseptically collected from the centre of each sample and diluted with 90 mL of sterile buffered peptone water (0.1% *w*/*v*) and homogenized in a stomacher (Lab Blender, West Sussex, UK) for 90 s. A series of 10-fold dilutions were prepared in sterile peptone water (0.1% *w*/*v*) (BPW, Biokar diagnostics, France) and inoculated in triplicate. *L. monocytogenes* enumeration was obtained after incubation in Oxford agar (Biokar Diagnostics BK110) supplemented with Oxford gelose (Biokar Diagnostics BS003) at 37 °C for 48 h. The natural microbiota of the control fresh cheese was also evaluated. Total mesophilic counts [21] were obtained after incubation in Plate Count Agar (PCA, Liofilchem, Teramo, Italy) at 30 °C for 72 h, the lactic acid bacteria counts [18] were obtained on Man Rogosa Sharpe (Oxoid CM0361, UK) at 30 °C for 72 h, and mould and yeast counts [22] were obtained after incubation in Chloramphenicol Glucose Agar (Oxoid CM0549, UK) at 25 °C for 3–5 days.

### 2.7. Data Analysis

All experiments were carried out in triplicate. The effect of the addition of myrtle and rosemary EO (plus control without EO) against *L. monocytogenes* and natural microbiota was studied by one-way analysis of variance. The differences (*p* < 0.05) among mean values were determined using the Tukey–Kramer test (SPSS 19.0, IBM, Armonk, NY, USA).

## 3. Results and Discussion

### 3.1. Chemical Composition of the Essential Oils

The EOs of myrtle and rosemary were analyzed by GC-MS. Results of their volatile composition are presented in Figure 1 and Figure 2. Twenty-five components represented 94.99% of the total of *Myrtus communis* EO while twenty-one components represented 86.62% of the total of *Rosmarinus officinalis* EO.

Myrtle EO was constituted mainly by hydrocarbon monoterpenes (α-pinene, 21.90%; myrtenil acetate, 19.82%; β-linalool, 11.44%) and the alcohol prenol (12.46%). Other reports indicated [23] α-pinene (35.20%), 1,8-cineole (17.00%), limonene (8.94%) and linalool (6.27%) or [24] α-pinene (46.9%), 1,8-cineole (25.20%) and absence of myrtenil acetate. The major constituents of rosemary EO were 1,8-Cineole (14.80%), β-pinene (9.40%), verbenone (9.15%), borneol (8.72%) and camphor (8.13%). Other works reported [25] 1.8-cineole (26.54%), α-pinene (20.14%), camphor (12.88%) and camphene (11.38%); [26] α-pinene (23%), camphene (7.6%), borneol (16%), camphor (4.5%), verbenone (9.4%) and borneol acetate (10.4%) or [27] camphor (22.4%), 1,8-cineole (13.2%) and α-pinene (10.8%). Variations of chemical compounds have been associated to several factors such us botanic variety, geographic origin or extraction method [28,29]. The antilisterial effect of myrtle and rosemary EO used has also been reported [30,31].

Also, some constituents such as 1,8-cineole, limonene, methyl-eugenol, α-pinene, α-terpineol, α-terpinolene and β-pinene were present in both EO, representing 44.61% and 39.76% of myrtle and rosemary EO, respectively. According to the chemical classification, myrtle and rosemary EO were mainly composed of monoterpenes (about 75% and 81%, respectively) and less than 1% of sesquiterpenes and phenylpropanoids compounds. 

### 3.2. Physical–Chemical Results in Control Samples of Fresh Cheese

Figure 3 presents the values of pH and a_w_ in control samples of cheeses. A reduction of pH over time was observed, showing pH values of 5.84 at the end of the storage of cheese.

Since pH is one of the most critical parameters regarding food safety and quality during cheese making, its determination is important to characterize cheeses due to its influence on texture, microbial activity and maturation. Additionally, there are chemical reactions catalyzed by enzymes derived from rennet and microbiota, which depend on the pH [32,33]. Milk has a pH of about 6.8, which means that, in terms of pH, it is an appropriate medium for the multiplication of most bacteria [34]. The reduction in the pH observed from 6.68 to 5.84 can correspond to the conversion of lactose into lactic acid by the action of lactic acid bacteria and can contribute to the prevention of the development of pathogenic bacteria [33]. The a_w_ values decrease over time from 0.98 in beginning until 0.94 at end of storage of control cheeses. The a_w_ is an important parameter for microbial development. Cheeses with higher a_w_ present a greater tendency to deteriorate or to support the multiplication of pathogenic and spoilage microorganisms [35]. 

Low levels of pH and a_w_ contribute to the safety of foods since they create stress conditions that hurdle the growth of a microorganism. However, the observed increase in *L. monocytogenes* counts during cheese ripening in control samples indicates the ability to survive at low pH and a_w_ levels [6]. Then, the results showed that other control strategies must be taken to reduce the growth of *L. monocytogenes* to guarantee the safety and quality of this type of cheeses [6,36].

### 3.3. Effect of Myrtle and Rosemary EOs on L. monocytogenes and Natural Microbiota of Fresh Cheese

The usage of EO as natural antimicrobial agents has been described in several food commodities [8,37,38] but the application of EO in cheese has been scarcely referred to [39]. Among them, the use of EO of black cumin [40], thyme [41], oregano [42], clove [43] or rosemary [44] have been referred but, to the best of the authors’ knowledge, this is the first report about the use of myrtle EO in cheese as an antimicrobial agent. The addition of myrtle or rosemary EO during the manufacturing of cheese (Table 1), revealed to be an effective method to control the growth of *L. monocytogenes* during the storage period (*p* < 0.01). 

As expected, counts of *L. monocytogenes* in cheese without EO, increased along the storage period to about 1.5 log CFU/g. Contrarily, *L. monocytogenes* counts were, on average from 1.23 log CFU/g to 1.74 log CFU/g lower in cheese made with myrtle or rosemary EO during the 28 days of storage period respectively. In cheese with added rosemary EO, *L. monocytogenes* counts increased 0.12 log CFU/g (about 2%) along the 28 days of storage (*p* > 0.05). However, a reduction of about 0.25 log CFU/g was observed during the last 14 days of storage.

Similar results were previously reported [45] in which the addition of rosemary EO in mozzarella cheese during ripening did not exert strong antilisterial activity, but retarded its growth. Despite the fact that *L. monocytogenes* counts on cheese made with myrtle EO were 0.5 log CFU/g lower than on cheese made with rosemary EO, no statistical differences were observed (*p* > 0.05) after means comparison by Tukey test along the storage period. Additionally, a decrease of approximately 0.2 log CFU/g was observed after 72 h of storage, opposite to what was observed in cheese with rosemary EO, in which *L. monocytogenes* counts increased within the same period. However, the behaviour of *L. monocytogenes* after this point was similar to those observed in cheese made with rosemary EO, reaching the highest count on day 14 of storage and decreasing afterwards. Thus, counts of *L. monocytogenes* increased about 25% in fresh cheese without the addition of EO 2% in those made with rosemary EO and decreased about 4% in cheese made with myrtle EO. However, a variation on *L. monocytogenes* counts along the storage period in cheese made with EO were not significant (*p* > 0.05) as indicated above. 

Use of myrtle or rosemary EO during cheese making improved the food safety of cheese (*p* < 0.01) against *L. monocytogenes* compared to the control batch. The anti-listerial effect of rosemary and myrtle EOs has been referred to in the literature [46,47], however, the use of these EO in our study only inhibits the growth, but does not have an effect in reducing the bacterial load. Additionally, other authors indicated that the main chemical compounds of rosemary and myrtle EO displayed a lower antilisterial effect in the food matrix than in vitro assays [48]. Regarding fat content, a higher inhibition of *L. monocytogenes* in low-fat soft cheese than in high-fat soft cheese has been reported [49]. Additionally, the negative effect of fat [50] decreases the antimicrobial activity of cinnamon and clove EO against *L. monocytogenes* in whole milk compared to skimmed milk. It has been suggested that high fat content increases the dispersion of the EO in the food matrix and decreases its contact with pathogens. Protein foods may be decreasing the antimicrobial effect of EOs as reported in beef extract and in fresh minced fish against *L. monocytogenes*. [51,52]. It has also been suggested that high levels of protein could decrease the interaction between EO and microorganisms due to the formation of a three-dimensional matrix of proteins that acts as a barrier [53,54] or by their hydrophobic properties [55,56]. This fact has been evidenced in another study [57], in which a total of 62% recovery yield of rosemary EO was observed from cheese samples after ripening. Since our cheese was made by pasteurized milk and clot is formed by addition of rennet (enzimatic coagulation), the high stability of the clot may act as a barrier in the diffusion of the EO, making its contact with *L. monocytogenes* difficult. 

Low pH may also influence the antibacterial effect of EO. Thus, the growth inhibition of rosemary and myrtle EO observed may be associated with an increase of the hydrophobicity of EO that facilitates the dissolution of the lipids present in the outer membrane of *L. monocytogenes*. Although *L. monocytogenes* can adapt to low pH levels as observed in the control group, the antilisterial effect of the EO may be enhanced by the pH conditions.

The effects of EO in total mesophilic counts (TMC), lactic acid bacteria (LAB), and mould and yeast (MY) counts, according to the time of storage are presented in Table 2. Total mesophilic bacteria increased along the storage period with initial counts at about 4 log CFU/g, and these data are in accordance as reported in the literature [45]. 

Variations on the initial counts of TMC are associated with the microbiological characteristics of raw milk as well as with the good hygiene practices during cheesemaking. The increase of about 8 log CFU/g in 14 days during ripening has also been reported [46,58]. The use of EO seems to influence the TMC (*p* < 0.001) however, after day 28 of ripening, similar values of TMC were observed between both control and EO-supplemented cheese. A similar growth pattern of TMC was described [46] in cheese made with oregano and rosemary EO. Although the addition of EO influences the TMC (*p* < 0.001), the difference in the *L. monocytogenes* levels in cheese prepared without EO was less than 0.1 log CFU/g. 

LAB represents the most important microbial group in cheese and is the major sources of enzymes responsible for ripening, throughout several basic mechanisms such as carbohydrate fermentation, conversion of milk proteins into peptides and free amino acids, catabolism of amino acids into aromatic compounds or hydrolysis of milk lipids into free fatty acids, among others. LAB remained stable (about 4 log CFU/g) during the first 3 days of ripening and increased until day 28 due to the fermentative activity of mesophilic microbiota [59], according to the decrease of the pH presented in Figure 3. However, other research [47] indicated that rosemary EO may exert a negative impact on LAB growth. Although the use of EO influenced the LAB counts (*p* < 0.01), samples added with EO displayed higher counts than control cheese. Other research [60] concluded that the addition of EO of lemon and thyme did not influence the LAB growth. No interference with native microflora as our study was also observed [61] against starter LAB culture using oregano EO and [47] using rosemary EO. Since LAB is mainly responsible for the development of the organoleptic characteristic of cheese, the absence of interference on LAB growth allows maintaining the identity characteristics of the cheese itself. However, [41] observed that, by adding thyme EO, a decrease in the counts of a starter culture of *L. cremoris* takes place. However, [61] no changes in the growth, acidifying activity and fermentative activity of native lactic acid bacteria were observed during cheesemaking of Argentinean cheese with oregano EO. Thus, the inhibition effect on EO must be previously assessed both in vitro and in real food products not only against foodborne pathogens but also against natural flora [33]. 

Cheese is usually an excellent substrate for mould growth, so some degree of mould manifestation is expected with time, usually being transposed on the consumer as a sign of spoilage. However, some mould species are used to produce specific types of cheese. 

Thus, the antifungal effect of rosemary and myrtle EO has been described in the literature [55,56]. However, information about the influence of EO during cheesemaking on the native fungal microflora is scarce. It has been observed [47] that the addition of rosemary EO during cheese making does not influence mould growth. Similar results were observed in our study in which moulds increased along the ripening period. The addition of myrtle or rosemary EO does not seem to influence mould growth. Thus, cheese made with rosemary EO displayed similar results as the control samples, in which mould counts increased after 3 days. Conversely, cheese made with myrtle EO showed over 2 log CFU/g of mould counts after 72 h of storage, when in comparison to the value of both the control and the cheese made with EO. 

As discussed above, the use of rosemary or myrtle EO improves the safety of cheeses, however, due to the aromatic nature of the EO, it may impact the sensory properties [16]. Although the sensory impact on consumer acceptance was not assessed in the present study, scarce research about the sensory evaluation of rosemary in cheese is available. Additionally, to the best knowledge of the authors, no information is available about the use of myrtle EO in cheese and its sensory impact. Regarding cheesemaking with rosemary EO, a classification of 6, 6, 8 and 6 points, in a 9-point hedonic-scale, was attributed to color, taste, odor and overall quality respectively [62]. Although the panelists considered the cheese neither likeable nor dislikeable, the impact on the aroma is evident, indicating the need for further studies on sensory impact.

## 4. Conclusions

In the present work, the application of rosemary or myrtle EO prevented the growth of *L. monocytogenes*. Since the absence of *L. monocytogenes* in 25 g in dairy products is mandatory before the food has left the immediate control of the food business operator, the use of EO may control a potential *L. monocytogenes* growth in case of cross-contamination along the food chain. However, the fact that myrtle and rosemary EO used in the study displayed a bacteriostatic effect indicates that the antimicrobial effect of EO used must be carefully assessed together with the physical and chemical characteristics of the food.

Moreover, the scarce influence of EO against the natural microbiota, total mesophilic and LAB allows maintaining the proprietary identity characteristics of the cheese itself during the cheesemaking process. Despite the improvement in food safety, the potential sensory impact of EO should be further addressed. 

## Figures and Tables

**Figure 1 foods-10-01106-f001:**
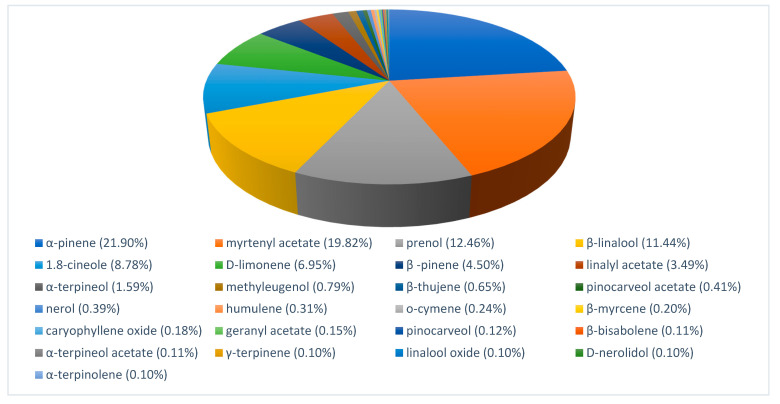
Rosemary essential oil chemical composition.

**Figure 2 foods-10-01106-f002:**
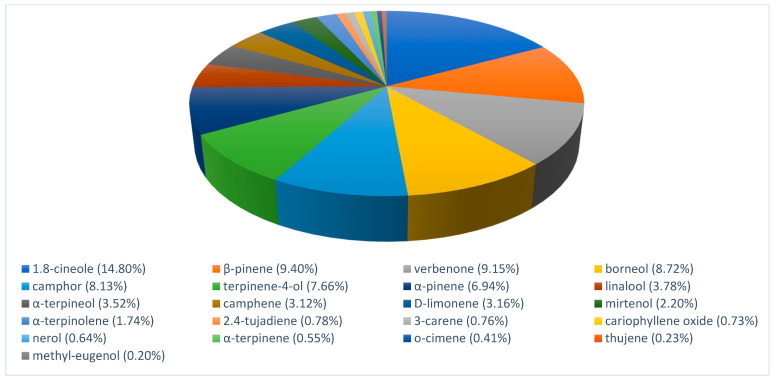
Myrtle essential oil chemical composition.

**Figure 3 foods-10-01106-f003:**
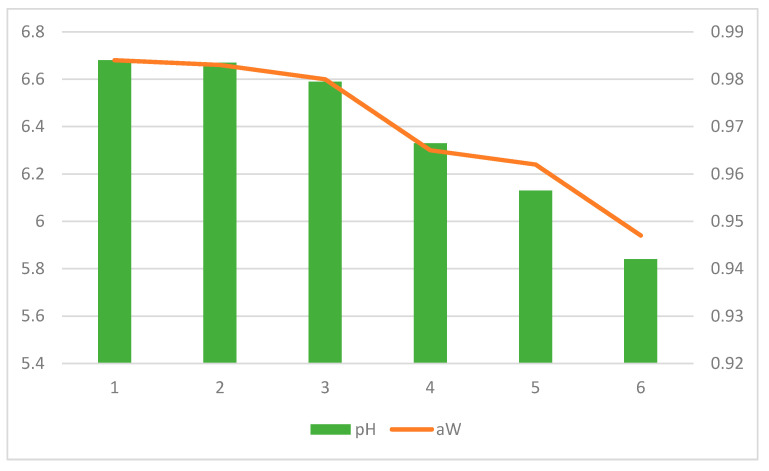
Values of pH (left axis) and a_w_ (right axis) in control samples stored at 8 °C during storage period.

**Table 1 foods-10-01106-t001:** Effect of essential oils (EO) on the survival of *L. monocytogenes* in cheese during ripening time.

Time	Control	Myrtle EO	Rosemary EO	Sig. (EO)
2 h	5.03 ^a^A ± 0.10	4.69 ^b^A ± 0.01	4.96 ^a^A ± 0.08	*p* < 0.01
24 h (1 day)	5.00 ^b^A ± 0.11	4.73 ^a^A ± 0.08	4.95 ^ab^A ± 0.08	*p* < 0.05
72 h (3 days)	5.63 ^a^B ± 0.04	4.54 ^b^A ± 0.47	5.11 ^ab^A ± 0.08	*p* < 0.01
168 h (7 days)	5.56 ^a^B ± 0.06	4.58 ^b^A ± 0.31	5.11 ^b^A ± 0.10	*p* < 0.01
336 h (14 days)	5.85 ^a^B ± 0.07	4.72 ^b^A ± 0.20	5.30 ^c^A ± 0.09	*p* < 0.001
672 h (28 days)	6.28 ^a^C ± 0.32	4.54 ^b^A ± 0.03	5.05 ^b^A ± 0.05	*p* < 0.001
Significance (time)	*p* <0.001	*p* > 0.05	*p* > 0.05	

Results are expressed as log CFU/g (mean ± standard deviation). In each row, means with different superscript letters differ significantly; In each column means with different capital letters differ significantly (*p* < 0.05).

**Table 2 foods-10-01106-t002:** The effect of essential oils (EO) on the natural microbiota in cheese, according to time of storage.

MO	Time	Control	Myrtle EO	Rosemary EO	*p* (EO)
TMC	2 h	4.44 ^ab^ ± 0.19	4.22 ^a^ ± 0.12	4.66 ^b^ ± 0.07	*p* < 0.05
	24 h (1 day)	6.37 ^a^ ± 0.29	4.73 ^b^ ± 0.32	5.99 ^a^ ± 0.02	*p* < 0.001
	72 h (3 days)	7.44 ^a^ ± 0.13	5.74 ^b^ ± 0.21	6.01 ^b^ ± 0.09	*p* < 0.001
	168 h (7 days)	8.16 ^a^ ± 0.14	5.99 ^b^ ± 0.04	7.63 ^a^ ± 0.51	*p* < 0.001
	336 h (14 days)	8.72 ^a^ ± 0.22	6.60 ^b^ ± 0.04	8.21 ^a^ ± 0.29	*p* < 0.001
	672 h (28 days)	9.58 ^a^ ± 0.25	7.96 ^b^ ± 0.25	9.47 ^a^ ± 0.11	*p* < 0.001
LAB	2 h	4.79 ^a^ ± 0.35	4.83 ^a^ ± 0.40	3.71 ^b^ ± 0.10	*p* < 0.01
	24 h (1 day)	4.82 ^a^ ± 0.15	5.54 ^b^ ± 0.19	5.25 ^b^ ± 0.05	*p* < 0.01
	72 h (3 days)	4.60 ^a^ ± 0.07	5.20 ^b^ ± 0.56	5.44 ^ab^ ± 0.40	*p* < 0.05
	168 h (7 days)	6.31 ^a^ ± 1.22	6.13 ^a^ ± 0.20	7.60 ^a^ ± 0.10	*p* > 0.05
	336 h (14 days)	8.52 ^a^ ± 0.17	7.09 ^b^ ± 0.43	8.28 ^a^ ± 0.23	*p* < 0.01
	672 h (28 days)	8.82 ^a^ ± 0.04	8.07 ^b^ ± 0.17	8.95 ^a^ ± 0.05	*p* < 0.001
MLD	2 h	3.01 ± 0.15	3.06 ± 0.11	3.06 ± 0.11	*p* > 0.05
	24 h (1 day)	3.07 ^a^ ± 0.09	4.33 ^b^ ± 0.31	3.16 ^a^ ± 0.02	*p* < 0.001
	72 h (3 days)	2.98 ^a^ ± 0.09	5.34 ^b^ ± 0.67	3.06 ^a^ ± 0.11	*p* < 0.001
	168 h (7 days)	4.49 ^a^ ± 0.08	6.37 ^b^ ± 0.19	4.31 ^a^ ± 0.27	*p* < 0.001
	336 h (14 days)	5.61 ± 0.07	6.35 ± 0.28	5.33 ± 0.66	*p* > 0.05
	672 h (28 days)	7.04 ± 0.32	6.87 ± 0.42	7.21 ± 0.18	*p* > 0.05

Results are expressed as log CFU/g (mean ± standard deviation). MO: microorganisms, TMC: Total mesophilic counts; LAB: lactic acid bacteria, MLD: moulds. a, b, values in the same line with different superscripts letters are significantly different (*p* < 0.05).

## Data Availability

The data that support the findings of this study are available from the corresponding author (Cristina Saraiva), upon reasonable request.

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
