# Peer review of "Antimicrobial Activity of Myrtus communis L. and Rosmarinus officinalis L. Essential Oils against Listeria monocytogenes in Cheese"

_foods, 2021, doi:10.3390/foods10051106_

Round 1

Reviewer 1 Report

Manuscript Revision

Title: Antimicrobial Activity of Myrtus communis L. and Rosmarinus officinalis L. essential oils against Listeria monocytogenes in Cheese.

Personally, I find the manuscript interesting. I appreciate the efforts of the authors to develop a new product with greater food safety by using essential oils with possible antimicrobial activity against common pathogens in the food industry such as Listeria monocytogenes. However, the manuscript has a series of drawbacks that need to be corrected. In the next lines, I will provide a description of the main problems encountered in the manuscript.

In the title, the authors had a mistake and forgot to erase the word title from the template.

As for the abstract, I miss a few introductory sentences. The abstract begins with the aim of the study and no information is given about the interest of the research carried out. Moreover, the name of the compounds should not be capitalized (the same error exists from line 201). Additionally, some aspects are not clear. For example, in line 27 it says “accounting for 44.61% and 39.76% of myrtle and rosemary EO”. This percentage includes all the mentioned compounds, is it specific to some of them or does it not refer to the same in both essential oils?

In some cases, it is necessary to improve the level of writing of the text. E.g., line 49-51.

Line 54. From the way it is written, it seems that this aspect is only fulfilled in Europe.

Line 55. In this line, the text loses cohesion. It should be a union to switch from pathogen problems in food industries to the use of essential oils. In addition, in the following lines, there is a disorganization of the text. It is necessary to improve the structure of the introduction. In addition, I consider this section too brief and lacking in information.

In the Extraction of Essential Oils section, it is necessary to indicate the amount of raw material used.

Line 87-89. Re-phrase.

Line 136. Information about the provider is missing.

Line 139. “(1/4000) was” instead of “(1/4 000) was”.

Line 167. Water activity is a common parameter. However, it is necessary to indicate the meaning of the acronyms.

Line 194. The reference to the table appears in another color. Put it in black like the rest of the text. Do the same with the rest of the table names.

Line 211. “The antilisterial effect of myrtle and rosemary EOs used has been reported elsewhere”. If the information is not going to be explained or explain in which studies is based on, it would be better to delete this sentence

Possibility of transforming table 2 into a figure.

Some parts of section 3.3 seem more appropriate for the conclusion section. E.g., line 304-310.

The conclusion section is too short. It can be completed with information present in section 3.3.

It would be interesting to work more on the tables, and the elaboration of figures with the results could be interesting. In this way, it would be easier to interpret the results and it would also be more visual. Leaving it in its current state it seems that the results were not thoroughly studied.

Reference is made to the fact that in this study, as well as in other previous studies, the possible applicability of these essential oils. However, no information is provided about the applicability of the proposal. These essential oils are characterized by being very aromatic, so it is expected that they will affect the organoleptic characteristics of the cheese. Has this change been valued? Does consumer acceptance vary?

FINAL REMARKS

Overall, the manuscript is well written and presents all the necessary information to replicate the experiment. However, the article must be studied in greater depth, being essential to improve the introduction and interpretation of the results. Therefore, I am suggesting MAJOR REVISIONS and RECONSIDERATION. The study should be improved.

Author Response

Reviewer 1´s comments

Personally, I find the manuscript interesting. I appreciate the efforts of the authors to develop a new product with greater food safety by using essential oils with possible antimicrobial activity against common pathogens in the food industry such as Listeria monocytogenes. However, the manuscript has a series of drawbacks that need to be corrected. In the next lines, I will provide a description of the main problems encountered in the manuscript

RE: Thank to the Reviewer for the revision which helped us to improve the quality of the manuscript. The entire manuscript was newly revised according the reviewer´ suggestions.

1) In the title, the authors had a mistake and forgot to erase the word title from the template.

RE: Amended as indicated.

2) As for the abstract, I miss a few introductory sentences. The abstract begins with the aim of the study and no information is given about the interest of the research carried out.

RE: The reviewer is right in his/her remark. An introductory paragraph has been added to the abstract section as requested.

3) Moreover, the name of the compounds should not be capitalized (the same error exists from line 201).

RE: The name of all chemical compounds across the manuscript were corrected as indicated.

4) Additionally, some aspects are not clear. For example, in line 27 it says “accounting for 44.61% and 39.76% of myrtle and rosemary EO”. This percentage includes all the mentioned compounds, is it specific to some of them or does it not refer to the same in both essential oils?

RE: The percentage 44.61% and 39.76% refer from the total of chemical compounds of each essential oil. The objective of this sentence is to highlight that a few chemical compounds are present in both essential oils in similar quantity. To avoid misunderstanding, the sentence was improved as indicated.

5) In some cases, it is necessary to improve the level of writing of the text. E.g., line 49-51.

RE: The sentence was improved as indicated.

6) Line 55. In this line, the text loses cohesion. It should be a union to switch from pathogen problems in food industries to the use of essential oils. In addition, in the following lines, there is a disorganization of the text. It is necessary to improve the structure of the introduction. In addition, I consider this section too brief and lacking in information.

RE: The introduction section was revised and improved as indicated.

7) In the Extraction of Essential Oils section, it is necessary to indicate the amount of raw material used.

RE: Amended.

8) Line 87-89. Re-phrase.

RE: the sentence was improved.

9) Line 136. Information about the provider is missing.

RE: Amended.

10) Line 139. “(1/4000) was” instead of “(1/4 000) was”.

RE: Amended.

11) Line 167. Water activity is a common parameter. However, it is necessary to indicate the meaning of the acronyms.

RE: The meaning of the acronym was indicated in the text.

12) Line 194. The reference to the table appears in another color. Put it in black like the rest of the text. Do the same with the rest of the table names.

RE: The colour was corrected along the manuscript as indicated.

13) Line 211. “The antilisterial effect of myrtle and rosemary EOs used has been reported elsewhere”. If the information is not going to be explained or explain in which studies is based on, it would be better to delete this sentence

RE: The sentence was modified as suggested.

14) Possibility of transforming table 2 into a figure.

RE: Table 2 was transformed into a figure as suggested.

15) Some parts of section 3.3 seem more appropriate for the conclusion section. E.g., line 304-310.

RE: The sentence indicated was moved to the conclusion section.

16) The conclusion section is too short. It can be completed with information present in section 3.3.

RE: See point 15.

17) It would be interesting to work more on the tables, and the elaboration of figures with the results could be interesting. In this way, it would be easier to interpret the results and it would also be more visual. Leaving it in its current state it seems that the results were not thoroughly studied.

RE: As suggested by reviewer, table 1 and table 2 were transformed into a graph. However, authors consider that conversion of table 3 and table 4 into a graph decrease the information provided for the reader as mean differences assessed by Tukey-Kramer test are not well represented into a graph.

18) Reference is made to the fact that in this study, as well as in other previous studies, the possible applicability of these essential oils. However, no information is provided about the applicability of the proposal. These essential oils are characterized by being very aromatic, so it is expected that they will affect the organoleptic characteristics of the cheese. Has this change been valued? Does consumer acceptance vary?

RE: Authors inform the reviewer that no sensory evaluation was studied, However, some information about sensory impact of rosemary EO in cheese was added to the text. As indicated in the text, no information (at the best knowledge of authors) is available about the sensory and antimicrobial efficacy of myrtle EO in cheese.

19) FINAL REMARKS. Overall, the manuscript is well written and presents all the necessary information to replicate the experiment. However, the article must be studied in greater depth, being essential to improve the introduction and interpretation of the results. Therefore, I am suggesting MAJOR REVISIONS and RECONSIDERATION. The study should be improved

RE: All the queries previously indicated were corrected as suggested by the reviewer. The manuscript was revised and improved throughout. Also, some tables were transformed into graph to improve the visual quality and the reader understand.

Reviewer 2 Report

The manuscript "Antimicrobial Activity of Myrtus communis L. and Rosmarinus officinalis L. essential oils against Listeria monocytogenes in Cheese" presents an interesting possibility of using essential oils as an antimicrobial substance in cheese.

Detailed comments:

line 1 - remove the word Title

line 31 - it is enough to enter the serial number of the strain once

The Introduction chapter is very reduced. It needs to be expanded to introduce the reader to the topic. The action of other EO on L. monocytogenes and the use of EO in cheese production can be presented (e.g. https://doi.org/10.1016/j.ifset.2017.09.020)

line 201-210 - there is no point in repeating the data from the table, please edit it.

The chapter Conclusion should be expanded to include the most important research results. 

Author Response

REVIEWER 2 COMMENTS

The manuscript "Antimicrobial Activity of Myrtus communis L. and Rosmarinus officinalis L. essential oils against Listeria monocytogenes in Cheese" presents an interesting possibility of using essential oils as an antimicrobial substance in cheese.

RE: Thank to the Reviewer for the revision which helped us to improve the quality of the manuscript. All the reviewer’ suggestions were taken into consideration.

Detailed comments:

1) Line 1 - remove the word Title

RE: Amended as requested.

2) line 31 - it is enough to enter the serial number of the strain once

RE: The serial number was indicated once. Following references were removed.

3) The Introduction chapter is very reduced. It needs to be expanded to introduce the reader to the topic. The action of other EO on L. monocytogenes and the use of EO in cheese production can be presented (e.g. https://doi.org/10.1016/j.ifset.2017.09.020)

RE: The introduction section was improved as indicated.

4) Line 201-210 - there is no point in repeating the data from the table, please edit it.

RE: Table was transformed into a graph.

5) The chapter Conclusion should be expanded to include the most important research results. 

RE: The conclusion section was improved.

Round 2

Reviewer 1 Report

Manuscript Revision

Title: Antimicrobial Activity of Myrtus communis L. and Rosmarinus officinalis L. essential oils against Listeria monocytogenes in Cheese.

The authors have carried out interesting work to improve the article. Despite this, some aspects still need to be improved. In the following lines, I explain the main problems to solve.

Line 53 – 56. These lines are a word-by-word repetition of what is written in the abstract. Rewrite it so that it is not the same.

Line 64. “surfaces, among others” instead of “surfaces among others”.

Line 65 – 68. Need to put in context. Those are practices that all food industries should have, not just dairy industries. Here he does not focus on the cheese and also takes a leap in the narrative that later does not follow. A possible solution might be:

Line 64: “others [6]. Therefore, due to the ubiquitous” instead of “others [6]. Due to the ubiquitous”

Line 69. “despite these measures, cheese is still a food susceptible to” instead of “Cheese is a food very susceptible to”.

Line 80 – 88. It would be advisable to unify both paragraphs.

Line 87. “in vitro” should be in italics.

Line 149. “As defined elsewhere” it is not scientific language. It would be more convenient to write something in the style as defined in previous studies.

Line 191. “water activity (aW))” instead of “water activity - aW)”.

Line 225. It would be interesting to put the images of the chromatograms obtained in the process.

Figure 3. Names are missing on chart axes.

Table 2. Does this table not enter in the previous sheet? There is a huge blank space.

Line 411. It is necessary to put the species from which the EO is extracted. As it is written, it seems to indicate that all essential oils have these properties.

Line 415. “EOs” instead of “EOS”.

FINAL REMARKS

Overall, the manuscript is well written and presents all the necessary information to replicate the experiment. Furthermore, the authors have carried out most of the suggested changes, having substantially improved the quality of the manuscript. However, it is still possible to find mistakes that need to be improved before the manuscript is eligible for publication. Therefore, I am suggesting MINOR REVISIONS. The study should be improved before publication.

Author Response

Reviewer 1´s comments

The authors have carried out interesting work to improve the article. Despite this, some aspects still need to be improved. In the following lines, I explain the main problems to solve.

RE: Thanks to the Reviewer for the revision which helped us to improve the quality of the manuscript. The manuscript was revised according the reviewer´ suggestions.

Line 53 – 56. These lines are a word-by-word repetition of what is written in the abstract. Rewrite it so that it is not the same.

RE: The paragraph was rewritten as indicated.

Line 64. “surfaces, among others” instead of “surfaces among others”.

RE: Amended.

Line 65 – 68. Need to put in context. Those are practices that all food industries should have, not just dairy industries. Here he does not focus on the cheese and also takes a leap in the narrative that later does not follow. A possible solution might be:

Line 64: “others [6]. Therefore, due to the ubiquitous” instead of “others [6]. Due to the ubiquitous”

Line 69. “despite these measures, cheese is still a food susceptible to” instead of “Cheese is a food very susceptible to”.

RE: The sentences were rephrased as indicated.

Line 80 – 88. It would be advisable to unify both paragraphs.

RE: Amended.

Line 87. “in vitro” should be in italics.

RE: Amended.

Line 149. “As defined elsewhere” it is not scientific language. It would be more convenient to write something in the style as defined in previous studies.

RE: Amended.

Line 191. “water activity (aW))” instead of “water activity - aW)”.

RE: Amended.

Line 225. It would be interesting to put the images of the chromatograms obtained in the process.

RE: Thanks to the Reviewer for the suggestion. However, authors consider that new figures (e.g. cake graphs) presented after the first review are more interesting to readers, since they displayed the amount of each chemical compound.

Figure 3. Names are missing on chart axes.

RE: Thanks to the Reviewer for the suggestion. Scale references of both axes are indicated in figure caption.

Table 2. Does this table not enter in the previous sheet? There is a huge blank space.

RE: Amended.

Line 411. It is necessary to put the species from which the EO is extracted. As it is written, it seems to indicate that all essential oils have these properties.

RE: Amended.

Line 415. “EOs” instead of “EOS”.

RE: Amended.

FINAL REMARKS

Overall, the manuscript is well written and presents all the necessary information to replicate the experiment. Furthermore, the authors have carried out most of the suggested changes, having substantially improved the quality of the manuscript. However, it is still possible to find mistakes that need to be improved before the manuscript is eligible for publication. Therefore, I am suggesting MINOR REVISIONS. The study should be improved before publication.

RE: All suggestions made by reviewer were carefully checked and corrected. The entire manuscript was newly revised and corrected.